# Niobium Base Superalloys: Achievement of a Coherent Ordered Precipitate Structure in the Nb Solid-Solution

**François Saint-Antonin [1,\*], Williams Lefebvre [2] and Ivan Blum [2]**

[1]   Grenoble Alpes University, CEA-Grenoble, LITEN, DTNM, L2N, 17 rue des Martyrs,
      38054 Grenoble CEDEX 09, France

[2]   Normandie University, UNIROUEN, INSA Rouen, CNRS, Groupe de Physique des Matériaux,
      76000 Rouen, France

\*   Correspondence: francois.saint-antonin@cea.fr; Tel.: +33-043-878-5717

**Abstract:** In a previous work, the chemical elements necessary for the achievement of Niobium base superalloys were defined in order to get a structure equivalent to that of Nickel base superalloys, which contain ordered precipitates within a disordered solid-solution. It was especially emphasized that precipitation hardening in the Niobium matrix would be possible with the addition of Ni. The remaining question about the design of such Niobium superalloys concerned the achievement of ordered precipitates in crystalline coherence with the Nb matrix i.e., with a crystalline structure equivalent to the Nb crystal prototype and with a lattice parameter in coherency with that of the Nb matrix. In order to reduce the trial/error experimental work, a reasoning based on various data for the achievement of coherency is presented. Then, starting from the Nb-Hf-Ni ternary alloy thus defined, this paper demonstrates that the precipitation of an ordered Nb phase within a disordered Nb matrix can be achieved with lattice parameter coherency between the ordered precipitates and the disordered matrix. The chemistry and the crystallographic structure of the precipitates were characterized using Transmission Electron Microscopy and Atom Probe Tomography. These results can help to conceive a new family of Nb base superalloys.

**Keywords:** Niobium superalloys; alloy design; ordered coherent precipitates; transmission electron microscopy; atom probe tomography

## 1. Introduction

The high-resistance of Nickel base superalloys is essentially due to the presence of precipitates, based on the ordered $Ni_3(Al, Ti)$ phase, named $\gamma'$, in a Ni-based solid-solution, named $\gamma$, or the matrix [1,2]. It should be emphasized that the Ni-based matrix and the $\gamma'$ phase have both a face-centred cubic (FCC) type structure and that there is a lattice parameter (near-)coherency between the two phases. Indeed, the lattice parameters at room temperature of Ni and $Ni_3Al$ are very close i.e., respectively 0.352 nm and 0.3567 nm. Depending on the nature and content of element within the Ni matrix and, within the $\gamma'$ phase, the level of the lattice parameter coherency between the ordered precipitates and the disordered matrix can change [3]. Indeed, due to the difference of the Coefficient of Thermal Expansion (CTE), the level of coherency between the two phases changes slightly with temperature. This CTE difference induces some volume constrains followed by possible relaxation, which is a function of volume fraction and size of the precipitates. The level of coherency at the precipitate/matrix interface (PMI) also depends on the size and on the morphology of the $\gamma'$ precipitates. This latter aspect can be understood with the following 'rule of thumb' description:

- (1) (planar) epitaxy occurs when a small quantity of atoms are disposed onto the surface of atoms having a different nature from the deposited one,
- (2) the large number of atoms imposes their crystal structure to the small amount of atoms present on their surface (epitaxy),
- (3) with the increase of the atom quantity, the atoms in contact with a surface, can adopt their own crystal structure (i.e., independently from the atoms they are in contact).

This mechanism extrapolated to 'volume epitaxy' can explain that loss of coherency can occur when growing precipitates reach a defined volume (if, of course, the precipitates have no crystal possibility to find any adjustment based on a near-equivalent lattice parameter making reference to the matrix crystal structure, whatever the crystal orientation). If the stable situation is incoherency at the interface, the passage from small coherent precipitates to large incoherent ones can lead to precipitate deformations leading to twins (after twisting).

These two preceding aspects plus the fact that the γ′ precipitates are included in the volume of the γ matrix, have different impact on the γ/ γ′ interfacial strain, which can be compensated (at least partly) by local chemical gradient, through diffusion mechanisms (during fabrication processes, heat-treatments or working conditions).

In Ni base superalloys, there are two main hardening mechanisms linked to the γ′ phase and to the crystalline coherency with the Ni solid-solution [4] (synthesized from [5]):

- the *order-hardening*, is "related to the possible creation of a surface of antiphase boundary (APB) when a dislocation of the disordered matrix cuts across the precipitates" [4] (p. 2141),
- the *stacking-fault hardening*, related to the "creation of a high energy surface" through the dissociation of perfect dislocations into two partials, which hinders dislocation climb and cross-slip mechanisms. In γ/γ′ alloys "calculations show that at room temperatures, the APB energy accounts for about 80% of the magnitude of the flow stress" [6] (p. 563).

The lattice coherency at the PMI allows the dislocations to shear both the matrix and the precipitates in a continuous way as dislocations remain in the same plane. Indeed, in incoherent PMI, the dislocation loops are left around the precipitates and/or, are deviated by multiple cross-slip mechanisms. The stacking of several dislocations loops around the precipitates leads generally to void formation at a side of particles giving rise to possible crack initiation [4] (see Figure 17 plus associated comments and reference). Moreover, the presence of dislocations loops leads to strain hardening, building of intermixed dislocation structures and finally, generating a forest of dislocations i.e., blocking the motion of additional dislocations induced by increasing stresses. The possibility of stress relaxation is thus strongly hindered, leading to limited material adaptation to further straining or stresses [7,8].

Due to the high melting point of Nb (~2475 °C), several Niobium base alloys with different structural reinforcement approaches, were evaluated for high-temperature and high-strength applications [9–11]. The three main directions in Nb alloy development were focused on:

- alloys based on the Nb-Ti-Al system (for instance [12,13]). The idea behind the addition of Al and Ti is to increase to oxidation resistance and to get a low-density alloy: various amounts of Cr, V, Mo, Hf, W, Zr, and/or Si are added in order to increase the oxidation resistance and the mechanical properties [13].
- alloys with a Nb matrix reinforced with silicides such as $Nb_3Si$ and/or $Nb_5Si_3$ (for instance [14–19]).
- alloys with reinforcement based on carbides (for instance [20–22]), sometimes, in association with silicides (for instance [23]).

The main points to improve in alloys based on the Nb-Si system are presented in [24], and a review about Nb base alloys reinforced with silicides or carbides is given in [25]. To my knowledge, none of the developed Nb base superalloys has been reported presenting a precipitate/matrix crystalline coherency.

Even non-coherent precipitates can lead to very high strength and creep resistance starting from a rather small volume fraction of these, it should be emphasized that large volume fraction (up to

~80% for Ni base superalloys) of ordered phase can be designed when there is a crystalline coherency with the matrix structure. Everything being equal, ordered phases can lead to higher structural thermal stability than disordered solid-solutions. Indeed, atomic diffusion mechanisms in ordered phase involved coupled jumps with, at least, atoms from second nearest-neighbour sites whereas in disordered solid-solution, diffusion mechanisms involved first nearest-neighbour atomic sites [26]. Moreover, atomic plane shearing induced by stresses involved much more complex dislocations structures in ordered phase than in disordered solid-solution (see superdislocation in [27], see also pair of dislocations and superlattice stacking faults in [4], and [28]), which explains the high temperature strength and creep resistance.

From the preceding points, it appears that crystalline coherency between the precipitates and the encasing matrix is an essential structural aspect for reaching high temperature mechanical resistance for Nb base superalloys.

The first part of the paper is dedicated to the presentation of the reasoning based on various data in order to reach the chemical composition for the design of an alloy with crystalline coherency between the precipitates and the matrix. In a second part, a Nb-Hf-Ni ternary alloy prepared by arc-melting was characterized with Transmission Electron Microscopy and Atom Probe Tomography, showing that crystalline coherency can be achieved even the initial Nb-Ni system does not seem to support this possibility.

## 2. Method: Alloy Design towards the Achievement of Coherency

This part is dedicated to the design description of an alloy showing crystalline coherency. The reasoning for reducing the trial/error experimental work starts from the description of the chemicals necessary for the achievement of a precipitate-matrix structure referring to a work where the question of crystalline coherency was not addressed. As Ni was proposed for the induction of precipitation, the Nb-Ni binary phase diagram is then described leading to the point that this system does not present crystalline matching between Nb and any referenced Nb-Ni phases. Several aspects are thus presented that can be used for targeting crystalline coherency: one of the solutions proposed is to add a third element to Nb and Ni, which enlarges the affinity with Ni: Hf offers a larger potential than Zr. The Nb-Hf-Ni phases are described leading to the idea that there are many phases with various crystallographic symmetry and different number of atoms per cell. Based on these aspects, it is concluded that the Nb-Hf-Ni system presents the potential to create crystalline coherency as there is a large 'crystalline versatility' among the different Hf-Ni phases (no Nb-Hf-Ni ternary phases were described in the literature). The last step of the reasoning is then focused on the choice of the composition of the Nb-Hf-Ni ternary alloy that was cast and structurally characterized.

### 2.1. Starting Aspects from the Paper Entitled "Potential Niobium Superalloys

Based on Hume–Rothery's rules, on phase diagrams and on diffusion properties, the chemical associations for the development of Nb base superalloys with a structure equivalent to that of Nickel based superalloys, were proposed in [29,30]. Table 1 reports the different elements associated to their role for the achievement of a superalloy type structure composed of a solid-solution strengthening, phase precipitations within grain boundaries, precipitation of a second phase in the matrix and strengthening of these precipitates.

It was proposed to use Ni for inducing the precipitation of an ordered phase in the Nb matrix [31,32]: a point remaining to be clarified was how to achieve an ordered phase, for the precipitates, with a crystal structure equivalent to that of the Nb matrix with, if possible, lattice parameter coherency between the precipitates and the matrix.

**Table 1.** Comparative composition between Ni and Nb superalloys (adapted and completed from [31]).

| Microstructural Function | Nickel Base Superalloys | Niobium Base Superalloys |
|---|---|---|
| Elements necessary for phase precipitation in grain boundaries | C, B | Rare earth metals: Y, Er, ... |
| Base element | Ni | Nb |
| Solid-solution strengtheners and oxidation resistant elements | Cr, Co, Fe, Mo, W, Re | Mo, W, Re, Al, Ta, Ti, V, Cr |
| Elements needed for the precipitate formation in the matrix | Al, Ti | Ni (Co) [1] |
| Elements for the strengthening of the precipitated phase in the matrix | Nb, Ta, V | Zr, Hf |

[1] Co was mentioned as a possible candidate, but not considered because of the shortage risk in cobalt supply.

### 2.2. Comments about the Nb-Ni System

Even the Nb-Ni binary system has been studied several times [33–39], there are still some uncertainties for some of the phase boundaries [39]. The compound closest to the Nb side is $Ni_6Nb_7$ (called μ) with a composition ranging from ~49.6 to ~54.5 at% Nb. The crystal structure is of hexagonal type i.e., hR13 (in the Pearson symbol): this ordered structure contains 13 atoms per unit cell, this is to be compared to the Body-Centered Cubic (BCC) or cI2 structure for Nb (2 atoms per unit cell). The μ phase, $Ni_6Nb_7$ belongs to the Topologically Close-Packed family (TCP), whose formation in Ni base superalloys " ... generally results in an embrittling effect owing to the brittle nature of the phase" [40] (p. 182) and [41]). The structure of the μ phase $Ni_6Nb_7$ was described with site occupancy as a function of the Nb content, within its range of existence [42]. This shows that the crystallographic structure of $Ni_6Nb_7$ can allow a slight variability both in composition and site occupancy, which can be interpreted as reflecting some *'crystal versatility'* in the atomic stacking.

In other words, an ordered structure based on an alternation of 2 atoms with the shift from one position to the next one would lead to a 100% probability to find atom A just after atom B, and then, there is a 100% probability to find A after B (and so on). On the opposite side, in solid-solutions, as there is no organized order between A and B (i.e., A and B are perfectly substitutable), the probability of presence for an atom B to be located just after atom A is well below 100%. The exact probability depends on the atomic % of the solute in the solvent and leads generally to probability of presence that ranges between few percent to few tenth of a per cent. Any number reflecting a probability of presence that is below 100% but well above the probability that is encountered in solid-solution, would correspond to a situation between highly ordered phase and lowly ordered phase: this situation can be considered as ordered phase with a certain level of *'ordered crystal with atomic versatility'* (which corresponds to a large domain of existence in phase diagrams).

In the Nb rich side, the compound $Nb_5Ni$ was also reported with a crystallographic FCC structure (cF96) [43]. This phase was achieved after annealing Nb-Ni alloys during 1000 h at 800 °C. The authors indicate that they did not get the $Nb_5Ni$ phase in the pure state but only in equilibrium with Nb [43]. It must be stressed that Nb and Ta have very similar chemical and crystallographic properties: the same authors indicates that the same type of phase ($Ta_5Ni$) with the same crystal structure (cF96) is observed in Ta-Ni alloys [43]. The $Nb_5Ni$ phase has never been reported again since [44,45].

The two phases $Ni_6Nb_7$ and $Nb_5Ni$ have a large number of atoms per unit cell (respectively 13 and 96) compared to the Nb matrix (2). As reference, Ni and $Ni_3Al$, as structures are respectively cF4 (four atoms per unit cell with one chemical element) and cP4 (four atoms per unit cell with two chemical elements), have the same number of atoms (4) per unit cell and their basic structure is cubic. Coherency necessitates the same basic crystallographic structure: this is not the case in the Nb-Ni system on the Nb side: $Ni_6Nb_7$ is hexagonal and Nb cubic.

As a hypothesis, it can be considered that an *'ordered crystal with atomic versatility'* is the result of an atomic difference that is not sufficiently marked. As a support to this idea, it should be recalled

that a solid-solution structure is achieved if the atomic size and electronegativity difference between the two atoms of the solid-solution, are under a certain limit [31] (for instance, see references to Hume–Rothery's rules). Moreover, ordered structures are encountered when the atomic size and electronegativity difference becomes large (Hume–Rothery's rules again). An ordered structure with a very large number of atoms can be considered as a structure that is not sufficiently 'organized'.

### 2.3. In Search of Coherency for Nb Superalloy

The design of Nb base ordered precipitates with a number of atoms per unit cell that is about the same than in the Nb solid-solution unit cell (cI2) was supposed to be achieved with the addition of a third chemical to $Ni_6Nb_7$ phase (hR13). The role of this third element is to enlarge the atomic size and/or electronegativity difference with Nb (compared to Ni). Expressed in another way, the third element to be added to Nb-Ni should 'strongly' induce and organize the formation of ordered precipitates having a simpler crystal structure than the μ phase $Ni_6Nb_7$. For that purpose, some of the rules and criteria used for the design of the Nb superalloys were used as a starting point for the choice of the third element [31,32]. Especially, the paragraph 'Strengthening of the second phase' [31] (p. 169), indicates that Zr and Hf could be used as a precipitate strengthener: they have also larger atomic size and electronegativity difference with Nb when Ni is the reference for the difference.

Moreover, the solubility of Ni in Nb in the solid state is rather low: it is about 4 at% Ni in Nb at 1290 °C and decreases with temperature to about 1 at% Ni at 400 °C. In order to shift from the μ phase $Ni_6Nb_7$ to a simpler structure, two criteria were chosen for the design of the ternary alloy:

- the 'affinity' between Ni and the third element should be high, at least, higher than the one between Nb and Ni, so as to favour the formation of a phase with Ni. With this affinity condition, as Ni is essentially an element of the ordered precipitates, the third element should be within the precipitates and not within the Nb solid-solution,
- the solubility of the third element in Nb must be very large in order, (1) to be statistically present everywhere within the Nb matrix and close by Ni and, (2) to avoid the formation of compounds with Nb. The second point leads to a weak 'affinity' between the third element and Nb.

Thus, the affinity between the 3rd element and Nb must be lower than its 'affinity' with Ni. The largest solubility of the 3rd element within Nb means that its atomic radius should be close to the Nb one. The term 'affinity' can be related to the electronegativity difference between two chemical elements, which can be related to a scaling of the atomic bond strength [46]. In the Nb rich side, the Hf-Nb phase diagram [47] shows a larger solid-solution than the Nb-Zr phase diagram [48]. Moreover, there is a larger number of ordered phases in the Hf-Ni system than in the Ni-Zr system [49–53]. The different ordered phases of the Ni-Hf system are stoichiometric compounds, or referring to phase diagram, they are line compounds (LC). There are two phases in the Ni-Zr system with non-stoichiometric ordered phase. With the existence of a domain of existence, non-stoichiometric ordered phases should offer more flexibility for obtaining a coherent structure at PMI but a large number of stoichiometric ordered phases with various crystal symmetry and different atoms per unit cell may give a large flexibility for reaching crystal volumetric coherency with the Nb matrix solid-solution.

### 2.4. The Nb-Hf-Ni System

No Nb-Hf-Ni ternary crystallographic phases nor any ternary phase diagrams were found in the literature and in crystallographic database (see for instance [45,54]). Three different Nb-Ni-Zr ternary phase diagrams at 800 °C are reported in [55]: even they give different phase boundaries, they can be used as a base for thoughts as there is a proximity between Zr and Hf. Two of these phase diagrams show that the Ni solubility limit in Nb-Ni-Zr seems to be larger than in Nb-Ni (at 800 °C). If this would also be the case for Nb-Hf-Ni, that would be favourable for the solubility of Ni and thus, for the design of alloys with large volume fraction of precipitates.

Table 2 reports the crystallographic phases in Nb-Ni and Ni-Hf binary alloys. The Nb-Hf system is a solid-solution with a BCC (β-HfNb) solid-solution on the Nb rich side: a metastable solid miscibility gap is reported leading to a hexagonal α-Hf precipitation in the lower temperature part of the phase diagram [47]. It must be pointed out that the stoichiometry of phases with Nb and/or Hf can be very sensitive to oxygen and carbon impurities.

**Table 2.** Crystallographic data of phases based on Nb, Ni and Hf (synthesized from the different references mentioned in the present paper). In Pearson notation [56], the first letter refers to the crystal family, the second letter to centring type, the number corresponds to the number of atoms in the unit cell. The asterisk * stands for unknown crystallographic characteristic. LC stands for Line Compound (or fixed stoichiometry).

| Phase | Crystal Structure (Pearson Notation) |
|---|---|
| Nb | cI2 |
| Ni | cF4 |
| α-Hf | hP2 (below 1743 °C) |
| β-Hf | cI2 (above 1743 °C) |
| $Ni_6Nb_7$ (~49.6 to ~54.5 at% Nb) | hR13 |
| $Nb_5Ni$ (metastable?) (LC?) | cF96 |
| $Ni_2Nb$ (LC) | hP24, hP12, cF24 |
| $Ni_3Nb$ (LC) | oP8 |
| $Ni_8Nb$ (LC) | tI36 |
| $Ni_5Hf$ (LC) | cF24 |
| $Ni_7Hf_2$ (LC) | m** |
| α-$Ni_3Hf$ (LC) | hR12 |
| β-$Ni_3Hf$ (LC) | hP40 |
| $Ni_{21}Hf_8$ (LC) | aP29 |
| $Ni_7Hf_3$ (LC) | aP20 |
| $Ni_2Hf$ (LC) | cF24 |
| $Ni_{10}Hf_7$ (LC) | oC68 |
| $Ni_{11}Hf_9$ (LC) | tI* |
| NiHf (LC) | oC8 |
| $NiHf_2$ (LC) | tI12 |
| $NiHf_3$ (metastable) | oC16 |

Moreover, several other metastable phases were also reported in the literature for the different binary phase diagrams (not indicated in Table 2). Incidentally, it should be also noticed that the two phases $Ni_2Hf$ and $Ni_2Nb$ are reported in crystallographic database [45], but are not present in any publication about the Hf-Ni and Nb-Ni binary phase diagrams. The structure of these two phase diagrams in the 66.6 at%-Ni region does not seem to be compatible with the presence of $Ni_2Hf$ and $Ni_2Nb$. Moreover, three different $Ni_2Nb$ crystallographic structures are reported [45]: one of the three is the same as $Ni_2Hf$.

### 2.5. Design of the Nb-Hf-Ni Ternary Alloy

Compared to the Nb (cI2), the phases reported in Table 2 are different for either the crystal symmetry or the number of atoms per unit cell. Even it was supposed that the large number of crystal phases offered by the Hf-Ni system would be the base for targeting coherency at the PMI, the chemical continuity between the precipitate and the matrix should rely on Nb atoms in order to favour crystal and chemical matching. Thus, substitution of Ni and/or Hf by Nb atoms within the precipitates should be allowed. The composition of the Nb-Hf-Ni ternary alloy was defined keeping in mind the following aspects:

- the maximum Ni solubility in Nb is ~5 at% (at about 1290 °C),
- the limit of the BCC crystal structure of Nb with Hf in solid-solution is ~10 at% Hf at 250 °C [47],

- the solubility of Ni in Hf is lower than 1–2 at% Ni [49]: it was supposed that the addition of Hf to Nb alloy would reduce the Ni solubility,
- the stoichiometry of the Ni-Hf ordered phase closest to the Hf rich side of the phase diagram is NiHf$_2$: the Hf content is two times the Ni one.
- any modifications induced by a chemical for the three binary systems were supposed to be kept on a linear basis for the ternary system. This assumption is questionable but due to the rather small additions of Ni and Hf, it can be considered for a first approach. This rule can be applied for an estimate of the reduction of Ni solubility in Nb with addition of Hf.

In order to reduce peripheral precipitations, which can be the result of an excessive amount of Hf and/or Ni, it was chosen to keep the Ni content below its maximum solubility, taking into account the effect of Hf addition to Nb.

The Ni content was chosen to be ~4 at% to keep the maximum solubilizing potential for possible heat-treatments. The Hf content was chosen to be ~6 at% in order to allow Nb substitution in NiHf$_2$ (or in any other Ni-Hf ordered phase prototype for the precipitates).

Some guidance for the occurrence and crystal structure of ternary compounds is described in [57]: this paper published in 1995 was read well after the design of the Nb-Hf-Ni ternary alloy (in august 1992). Nevertheless, it can be concluded that if crystal structures for binary compounds can be deduced with a relatively high confidence, crystal structure prediction for ternary compounds is more difficult [57]. The same type of comment can be expressed for the calculation of ternary phase diagrams compared to binary ones.

## 3. Materials and Experimental Methods

About 300 g of the Nb-Hf-Ni ternary alloy was arc-melted with a tungsten electrode under an argon atmosphere. The purity was better than 99 at% for Nb (the main impurity is Ta), better that 97 at% for Hf (the main impurity is Zr, typically about 2.5 at%), and better than 99.5 at% for Ni. The secondary impurities in Nb and Hf are oxygen, carbon and refractory elements such as Mo, W, Ti (typically below few hundreds of ppm) and Ta also in Hf. It should be pointed out that regarding the electronegativity difference between on the one hand, Nb, Hf and their impurities (Ta, Zr, W, Mo, Ti) and on the other hand, Hf has the largest electronegativity difference (or affinity) with Ni. Thus, even the refractory impurities can represent a few percent, Ni should form mostly precipitates with Hf. Moreover, at these concentration levels, these refractory elements remain in solid-solution: they can also participate to the crystalline matching between the precipitates and the matrix. Even the purity of Hf is the lowest compared to Nb and Ni, considering a Zr amount of 2.8 at% in Hf, the Zr content introduced in the alloy with 5.8 at% Hf is about 0.16 at%. The quantity of the other impurities brought into the alloy with 5.8 at% Hf is rather very small. Tantalum which is the major impurity in Nb, has chemical and structural properties very close to Nb and should not have a strong impact on the precipitate structure (Ta is in solid-solution).

After arc-melting, the cooling step of the alloy was not registered: but a gentle cooling was performed allowing the precipitation. The composition of the alloy is: 90.3 at% Nb, 5.8 at% Hf, 3.9 at% Ni (or 86.9 wt% Nb, 10.7 wt% Hf, 2.4 wt% Ni). The properties and phase structures with Nb and Hf are sensitive to oxygen and carbon content. The carbon content measured by Atom Probe Tomography (APT) is about 150ppm (precision ± 0.001ppm). No tungsten (from the W electrode) was detected with APT. The oxygen content was not measured. No Mo was detected with APT. Zr and Ta should have appeared in the same mass position than Hf but cannot be seen as the Hf APT pic is stronger.

No specific heat treatment was performed after arc-melting. This alloy was conceived as a demonstrator for the achievement of crystalline coherency and for a validation of the reasoning approach for that purpose. The goal of this work was first to observe the structure in that state even it may be far from any stabilized structure and then, to define, in a second step, heat-treatment to promote precipitation growth or stabilization. If crystalline coherency is observed with small precipitates, it can

be lost during their growth. The information about the level of crystal matching is a starting point for further studies.

### 3.1. Specimen Preparation by Focussed Ion Beam (FIB) Milling

The specimens were prepared using the standard lift-out and mounting method [58] using a Plasma Focused Ion Beam "Helios G4 PFIB CXe" (Thermo Fisher Scientific, Eindhoven, Netherlands). A sample was first extracted from the Nb-Hf-Ni substrate by ion milling, using 30 kV $Xe^+$ ions. It was welded to a micromanipulator using ion beam induced deposition of platinum, and sections of the sample were welded in the same way on tungsten needles. These specimens were subsequently sharpened with the $Xe^+$ ion beam, using the annular milling method [58]. In order to minimize the thickness of the final damaged surface layer, the needles were then milled over ~100 nm using a 12 kV $Xe^+$ ion beam. Using this method, Nb-Hf-Ni needles with a radius of curvature below 50 nm could be obtained (Figure 1).

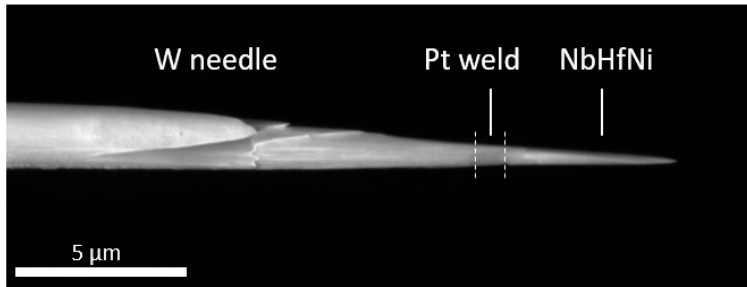

**Figure 1.** Scanning Electron Microscopy image of a Nb-Hf-Ni specimen after annular milling by FIB.

### 3.2. Atom Probe Tomography

Samples were analysed by Atom Probe Tomography (APT). This technique was selected because of its ability to analyse the chemistry and the morphology of precipitates at the atomic level. The experiments were carried out on a LEAP 4000 HR device from CAMECA (Gennevilliers, France). The experiments were performed at 60K with a flux of 0.25 ions/S/$nm^2$. The reconstruction procedure and analysis were conducted with the software package IVAS® (CAMECA, Gennevilliers, France). The cluster search algorithm, which is the maximum separation distance algorithm, was applied based on the distribution of the 8th nearest neighbour distances. A maximum separation distance of 0.8 nm was identified as relevant according to the comparison of experimental data to randomized data. A minimum amount of 35 solute atoms in clusters was set to remove all artificial clusters eventually detected in the solid solution. The reader can find more details about the methodology in reference [58].

### 3.3. Transmission Electron Microscopy

High Resolution Transmission Electron Microscopy (HR-TEM), Scanning Transmission Electron Microscopy (STEM) and Energy Dispersive X-Rays Spectrum (EDX) for elemental mapping, were carried out using a TECNAI OSIRIS (Thermo Fisher Scientific, Eindhoven, Netherlands) transmission electron microscope operated at 200 kV, equipped with a 4K GATAN camera.

The thin foil for TEM observation was prepared using classical mechanical polishing and ionic thinning machine. The thickness of the observed region is typically of several tenths of nm. Several thin foils were extracted from different parts of the ingot and showed similar TEM/STEM-EDX results about the structure and the chemistry of the precipitates. No other means were used to check homogeneity.

## 4. Experimental Results

### 4.1. Transmission Electron Microscopy Characterization

A single matrix dislocation shearing an ordered coherent phase creates an antiphase domain boundary (APB): this APB is removed by a second dislocation [59] (p. 334). In order to reduce APB, which has a high surface energy, dislocations are thus paired [59] (pp. 334–343). In Figure 2, there are clearly several pairs of dislocations with two configurations:

- (a) relaxed dislocations pairs corresponding to the shearing of the matrix; there is not any APB formed as the matrix is a disordered solid-solution, the two dislocations of the pairs are rather far apart (relaxed),
- (b) pinched regions, which are segments of strongly coupled dislocation pair, corresponding to the shearing of ordered coherent precipitates; this geometrical configuration reduces the size of the anti-phase domain boundary during shearing of the ordered precipitates [59] (p. 336). Between these pinched regions, the dislocation pair is relaxed.

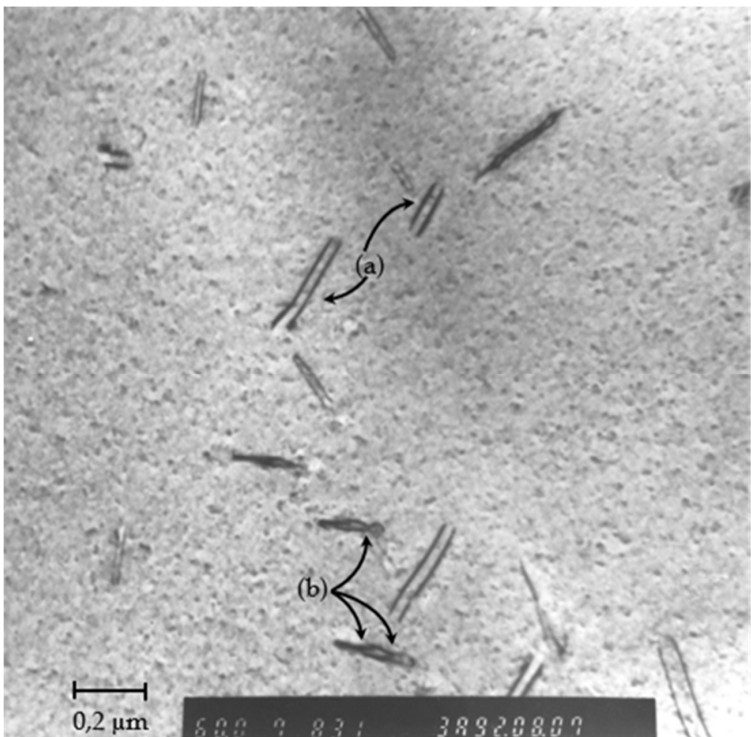

**Figure 2.** Pairs of dislocations, the pairing of dislocations is the result of the presence of coherent PMI. Legends: (**a**) two relaxed dislocation pairs (in the Nb solid-solution), (**b**) the arrows point towards the pinched regions of dislocation pairs, corresponding to strongly coupled segments of the dislocations. This image was obtained with a Hitachi MET at University of BC, Vancouver, Canada.

This image (Figure 2) is one of the first obtained on this alloy (the image is dated 07/08/92): in the frame of the dislocation theory, this is a real proof for coherency at PMI in this Nb-Hf-Ni ternary alloy. Indeed, as dislocations are geometrical defects at the atomic scale, except high resolution TEM, there is no better resolution for the proof of crystal coherency at PMI and of the achievement of equivalent crystal structure between the matrix and the precipitates.

Figure 3a is a diffraction pattern of the alloy: there are superlattice spots, which correspond to the ordered precipitates. The selection of one superlattice spot leads to the lighting up of the precipitates (Figure 3b).

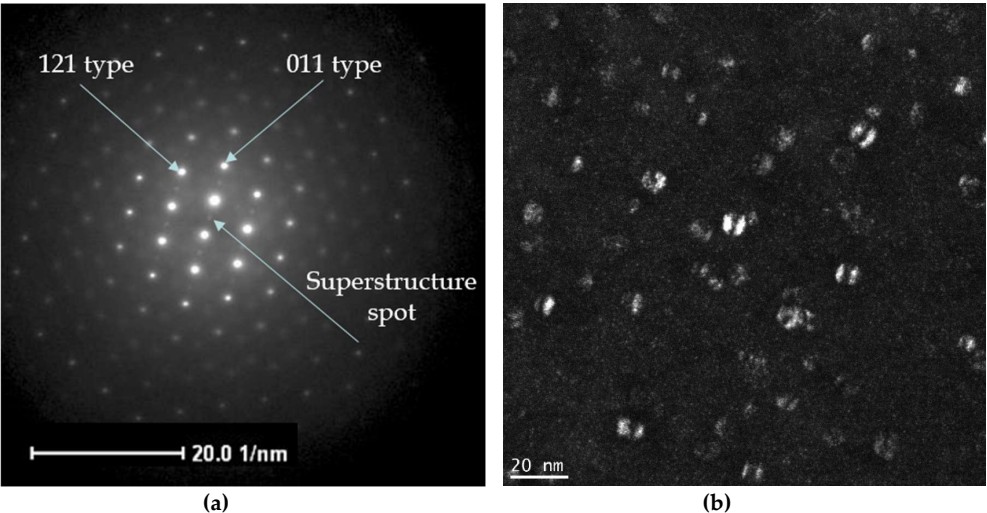

**(a)**          **(b)**

**Figure 3.** (**a**) BCC diffraction pattern with weaker superlattice spots. The orientation of the plane is (311). Brighter spots correspond to the Nb base BCC diffraction pattern plus precipitates, weaker spots only to the diffraction pattern of the precipitates. (**b**) Image of the particles based on the selection of a superlattice spot (dark field image). The two-hemispheric contrasts associated to each particle correspond actually to the strain field around the precipitates. The size of the precipitates is typically below 15 nm.

The precipitate sizes are comprised between 5 and 15 nm as suggested by the HR-TEM image presented in Figure 4. As the foil thickness is much larger than the precipitates, even the EDX maps (Figure 5) show the Hf and Ni higher concentration associated to precipitates and as there are also Hf and Ni within the matrix, it is not possible to extract an unambiguous chemical composition especially for crystallographic phases in the Nb-Hf-Ni system that are not referenced. Figure 4 shows that the interface between the precipitates and the matrix is not clearly defined, and that there is atomic plan continuity between the matrix and the precipitates. This aspect may be linked to non-stabilized precipitates. Further understanding of the matrix-precipitate structure is based on Atom Probe Tomography.

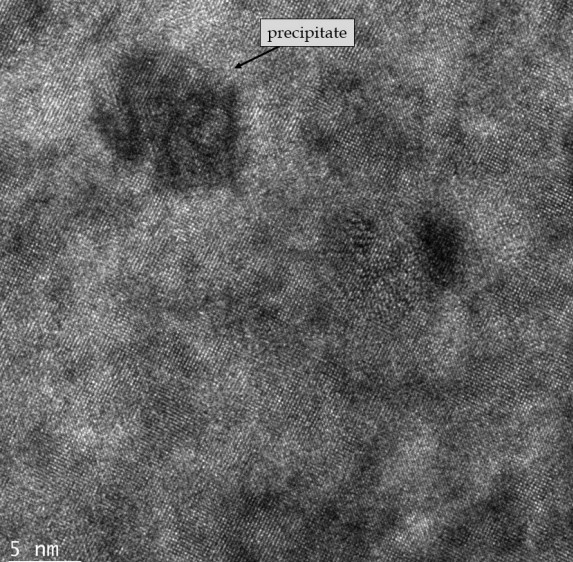

**Figure 4.** HR-TEM image of precipitates. The darker part of the image with circular shape are precipitates: the size is comprised between 5 and 15 nm. The interface between the precipitate and the matrix is not clearly defined and there is atomic plan continuity between the matrix and the precipitates.

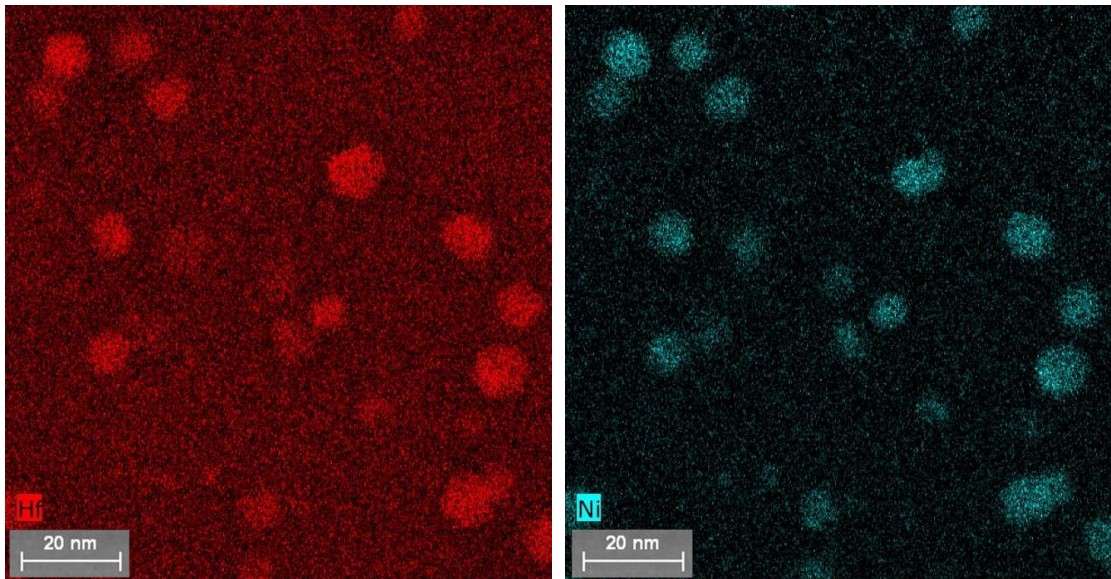

**Figure 5.** STEM-EDX map of Hf and Ni. The maps were treated using the ESPRIT software (Bruker Company) in order to minimize the noise. There is a clear concentration of Hf and Ni corresponding to the precipitates (5–15nm), but their chemistry is superimposed with the matrix chemistry as they are in the volume of the thin TEM foil (80–100nm).

## 4.2. Atom Probe Tomography Characterization

Two specimens of the alloy with different sizes were analysed. The corresponding reconstructions are shown in Figure 6. Without any data filtering, precipitates can unambiguously be distinguished. Meanwhile, weak local solute enrichments in the solid solution are visible. These objects appear more diffuse in the reconstruction and no clear interface can be identified between them and the matrix. These weak solute enrichments are later defined as clusters. These clusters are likely to correspond to the nuclei of the classical nucleation theory. Observing clusters and precipitates simultaneously indicates that the material is analysed in a state where nucleation and growth stages overlap.

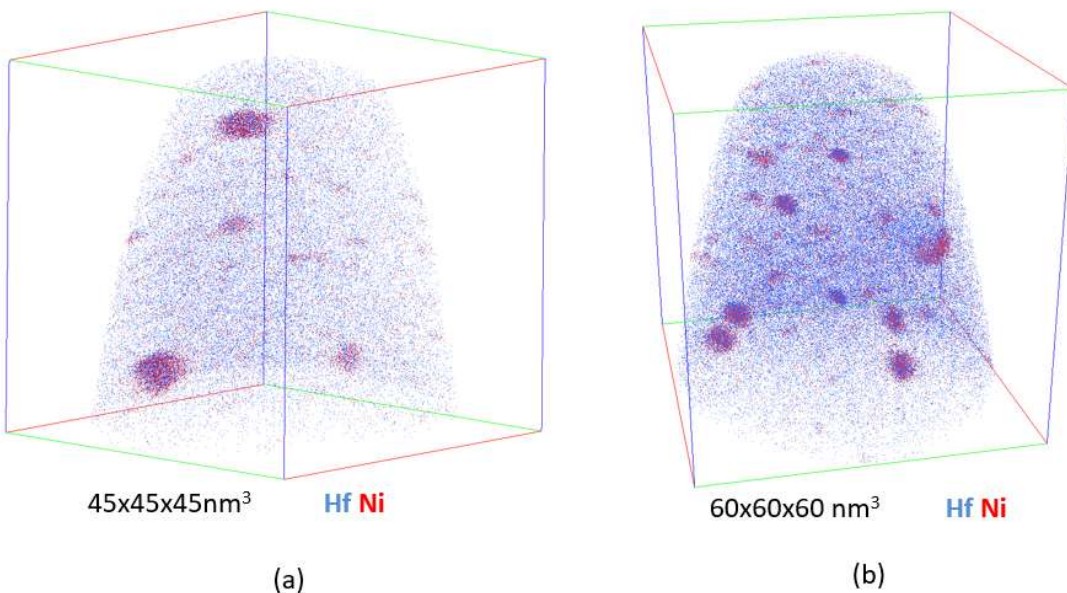

**Figure 6.** (**a**,**b**) are two distinct APT reconstructions obtained by two separate analyses performed on the alloy. Only Hf and Ni atoms are displayed. The two figures show particles with a rather well defined boundary (i.e. precipitates) and small clusters with more tenuous boundary.

Isosurfaces of Hf concentrations were calculated in order to identify interfaces between precipitates and the Nb matrix. These isosurfaces do not allow to identify the weak local solute enrichments, which are later investigated using a cluster selection algorithm. Figure 7a shows 8 precipitates. Concentration profiles plotted across the interface of two precipitates show similar trends (Figure 7b,c). Precipitates appear to consist in a diffuse interface across which the Nb composition decreases linearly. The precipitate core is almost free of Nb, whereas concentration of Ni and Hf seems to remain at constant concentration. The Hf/Ni relative concentration within the precipitates is comprised between 60/40 and 50/50 (or a Hf/Ni ratio between 1.5 and 1).

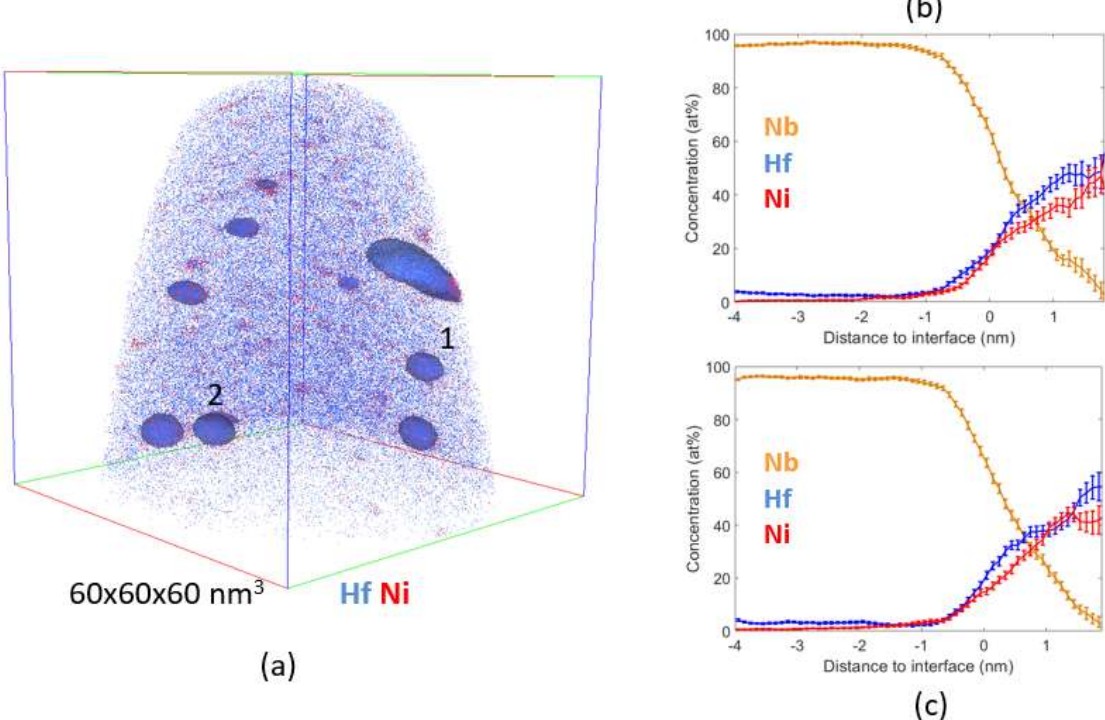

**Figure 7.** (**a**) Reconstruction of Figure 6b wherein isosurfaces were displayed for a concentration of Hf equal to 19%. This value was set empirically in order to allow the calculation of concentration profiles across these interfaces. (**b**,**c**) correspond respectively to the proximity histogram calculated along interfaces for precipitate 1 and 2 as shown in (**a**). In (**b**,**c**), the precipitate is on the left and the matrix on the right. The Hf/Ni relative concentration within the precipitates is comprised between 60/40 and 50/50.

Alongside with the precipitates, there are clusters corresponding to enriched solute and are interpreted as germs before precipitations. Clusters and precipitates were simultaneously analysed using a cluster selection algorithm (Figure 8). The local enrichments visible without any filtering of the data are now imaged and reveal a high density of small size clusters. Most of clusters contain less than 300 atoms (including Ni, Hf and Nb). The cluster diameter is less than 2 nm on average. The ratio of concentration of solute elements was calculated and displayed as a function of the number of atoms per cluster (Figure 8b). This statistical analysis reveals that the smaller the cluster, the larger the Hf concentration. As for the absolute Hf and Ni fractions in clusters: they are displayed as a function of the number of atoms per cluster in Figure 8c.

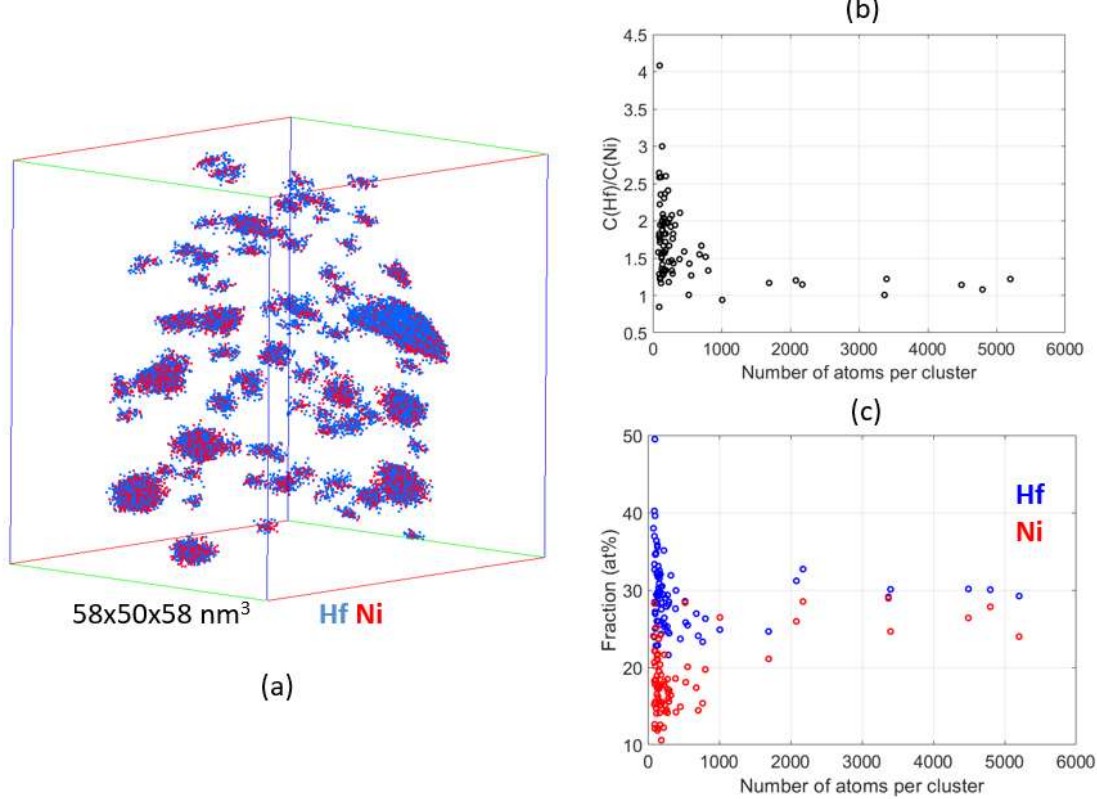

**Figure 8.** (**a**) Result of the cluster selection algorithm applied to the reconstruction of Figure 6b. (**b**) Concentration ratio Hf/Ni in clusters, as a function of the number of atoms in the cluster. (**c**) Hf and Ni fraction in clusters as a function of the number of atoms in the clusters.

It should be pointed out that, as for the precipitates, the Hf/Ni ratio for large precipitates is comprised between 1.5 and 1 (Figure 8b): clusters (enriched in solute) appear thus clearly as germs for the formation of precipitates as they grow in size.

## 5. Discussion

The alloy, whose composition is 90.3 at% Nb, 5.8 at% Hf, and 3.9 at% Ni, shows a typical superalloy structure composed with a solid-solution (or matrix) containing ordered coherent precipitates. The precipitates have a diameter comprised between 5 and 15 nm. The TEM image shows dislocation pairs. Moreover, there are strongly coupled dislocation segments of the pairs, which is a proof for crystal coherency at the precipitate/matrix interface, and a signature of an ordered precipitate structure with a crystal type derived from that of the matrix. STEM-EDX images show that the precipitates are enriched in Hf and Ni compared to the matrix chemistry. Due to the size of the precipitates (typically 5–15 nm) compared to the thickness of the observed region (80–100 nm), the determination of the exact chemistry of the precipitates was approached with Atom Probe Tomography. The use of APT is especially necessary when there is not a drastic chemical change at the interface between the precipitates and the encasing matrix. Indeed, APT shows that there is a diffuse interface between the matrix and the precipitates. Across the interface, the Nb content decreases from the matrix towards the precipitate cores within 2–3 nm: the precipitate core is almost free of Nb. Moreover, the Hf/Ni relative concentration within the precipitates is comprised between 60/40 and 50/50 with an ordered cubic structure in coherence with the matrix (cI2).

Alongside the precipitates, a large number of solute rich clusters, less than 2nm in diameter, are identified. In the present work, cluster sizes are accounted for based on the number of atoms associated to each cluster using the cluster search algorithm. Considering the number of atoms only does not allow neither a direct evaluation of metric size nor of the morphology of each cluster. This information

could be easily derived using some other algorithms. Nevertheless, insofar as cluster size is in the range of 1 nm, such information is actually deteriorated by the anisotropic spatial resolution of APT and may not reflect the actual size of clusters present in the material. It should also be mentioned that the detector efficiency of the APT used in this study is about 37%. This means that, on average, 37% of atoms are detected whatever species are considered. Consequently, the amount of atoms associated to each cluster is underestimated if one wants to get a direct comparison to the amount of atoms per cluster in the material.

Statistical analysis reveals that, as their size increases, they tend to contain similar amounts of Ni and Hf atoms. However, the smaller the clusters, the larger the Hf content. The clusters are identified as the nuclei of the new phase. Observing a large number density of clusters with precipitates is an indication that the material was analysed in a state where nucleation and growth stages of the new phase are overlapping. It hence appears that, as they grow, clusters initially rich in Hf tend to absorb Ni to reach a nearly equivalent content of solute atoms, in agreement with the ratio measured in the precipitates of the new phase.

As a final comment, even the cooling of the alloy after arc-melting was not recorded, the presence of both solute-rich clusters and precipitates offered the chance to have two states of the precipitate formation.

**Author Contributions:** Data curation, Formal analysis, Investigation, Visualization, Writing—original draft, F.S.-A. and W.L.; Methodology, F.S.-A., W.L. and I.B.; Supervision, F.S.-A.

**Funding:** the APT experiments were financially supported by CNRS CEA "METSA" French network (FR CNRS 3507). The other parts of the work received no external funding.

**Acknowledgments:** The authors acknowledge financial support from the CNRS-CEA "METSA" French network (FR CNRS 3507) on the plateform IRMA Groupe de Physique des Matériaux (Normandie Univ., UNIROUEN, INSA Rouen, France). One of the author, F. Saint-Antonin, wishes to address special thanks to Prof. Alec Mitchell: this Nb-Hf-Ni alloy was designed at the end of my post-doc (04/1991 – 08/1992) at the Department of Materials Engineering, University of British Columbia, Vancouver, Canada. The alloy was kindly arc-melted by the RMI Titanium Company, Ohio, USA: I would like to thank Mr Bertea for this fulfilment. The TEM image of dislocation pairs (Figure 2) was obtained on a thin foil performed during my post-doc (june-july 1992) at the National Center for Electron Microscopy, Lawrence Berkeley National Laboratory, Berkeley, California, USA. After a change of my position at CEA, this has been only since year 2012, that I have had the chance to use adapted Transmission Electron Microscope in order to begin the understanding of the precipitate structure of this Nb-Hf-Ni alloy. I would like to thank Dominique Delille (FEI at that time and now, Thermo Fisher Scientific) and Dominique Lafond (CEA/LETI) for the good training on this new TEM generation. Moreover, I would like to thank CEA/LITEN for the time I had for finishing this work and get a clearer idea about the bi-phased structure resulting from the composition upon which 'I bet on' more than 25 years ago. Finally, I always really appreciate the easy access to the Nano-characterization PFNC-MINATEC platform (CEA-Grenoble).

**Conflicts of Interest:** The authors declare no conflict of interest.

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
