# Peer review of "Niobium Base Superalloys: Achievement of a Coherent Ordered Precipitate Structure in the Nb Solid-Solution"

_crystals, doi:10.3390/cryst9070345_

Round 1

Reviewer 1 Report

The authors present the rational of their decision about selection of a new alloy, and then present the results of TEM-STEM and APT on the mentioned alloy.

The results are well described, but a lack of information about the starting material can be noted.

No information about the total mass of the cast material, or eventual preliminary investigation about cast homogeneity or other features, or XRD analyses.  Specimen are considered possibly taken form similar zones inside the cast. The specimen were observed in a as cast condition, after arc melting, without any other specification. So possibly in a rather fat cooled condition. It is quite normal to have a population of precipitates, if precipitation is allowed during cooling.

The considered themomechanical condition is rather distant from possible service condition, and the observed precipitates could be also a metastable phase. No comments on these points were reported.

Possibly additional consideration about the limit of APT on resolution should be mentioned when considering cluster sizes.

Discussion is extremely short compared to introduction. The introduction itself contain a lot of information about possible phases, list of binary phases ect, that are not later considered in the discussion or result presentation.

The content of the paper is good, but I suggest a re-writing in order to better relate the content of the introduction to the experimental part of the work.

Please pay attention to english language, and editing.

Examples:

-        Page 1-Line 25 capital letters for Transmission Electron Microscopy

-        P1 L31: …in a Ni based solid-solution matrix, named G.

-        P2 L58: remove ‘,’.

-        P2 L89; replace ‘synthesis’ with ‘review’

-        P4 L161 hR13

-        P12 L352, two specimen of the alloys were analysed, with different size.

-        P12 L358: no specific heat treatment was performed, so avoid reference to an applied heat treatment’.

-        P12 L354: A sentence is present twice. (‘Meanwhile…’).

-        P17 L540 Crystalline…

Author Response

The authors present the rational of their decision about selection of a new alloy, and then present the results of TEM-STEM and APT on the mentioned alloy.

The results are well described, but a lack of information about the starting material can be noted. No information about the total mass of the cast material, or eventual preliminary investigation about cast homogeneity or other features, or XRD analyses. Specimen are considered possibly taken form similar zones inside the cast. The specimen were observed in a as cast condition, after arc melting, without any other specification. So possibly in a rather fat cooled condition. It is quite normal to have a population of precipitates, if precipitation is allowed during cooling. The considered thermomechanical condition is rather distant from possible service condition, and the observed precipitates could be also a metastable phase. No comments on these points were reported.

In ‘Materials and Methods’, comments about the starting materials, homogeneity (gained by TEM analysis with several thin foils) were added. There was no preliminary investigations such as suggested by reviewer (this point has been added in paragraph 3.3). A comment was added about the cooling following the suggestion by the reviewer (to be noted: it was already specified that the cooling record was not available). Justification about the choice ‘no heat treatment after arc-melting’ was added.

The text is now:

“Materials and experimental methods

About 300g of the Nb-Hf-Ni ternary alloy was arc-melted with a tungsten electrode under an argon atmosphere. The purity was better than 99at% for Nb (the main impurity is Ta), better that 97at% for Hf (the main impurity is Zr, typically about 2.5at%), and better than 99.5at% for Ni. The secondary impurities in Nb and Hf are oxygen, carbon and refractory elements such as Mo, W, Ti (typically below few hundreds of ppm) and Ta also in Hf. It should be pointed out that regarding the electronegativity difference between on the one hand, Nb, Hf and their impurities (Ta, Zr, W, Mo, Ti) and on the other hand, Hf has the largest electronegativity difference (or affinity) with Ni. Thus, even the refractory impurities can represent a few percent, Ni should form mostly precipitates with Hf. Moreover, at these concentration levels, these refractory elements remain in solid-solution: they can also participate to the crystalline matching between the precipitates and the matrix. Even the purity of Hf is the lowest compared to Nb and Ni, considering a Zr amount of 2.8at% in Hf, the Zr content introduced in the alloy with 5.8at%Hf is about 0.16at%. The quantity of the other impurities brought into the alloy with 5.8at% Hf is rather very small. Tantalum which is the major impurity in Nb, has chemical and structural properties very close to Nb and should not have a strong impact on the precipitate structure (Ta is in solid-solution).

After arc-melting, the cooling step of the alloy was not registered: but a gentle cooling was performed allowing the precipitation. The composition of the alloy is: 90.3 at% Nb, 5.8 at% Hf, 3.9 at% Ni (or 86.9 wt% Nb, 10.7 wt% Hf, 2.4 wt% Ni). The properties and phase structures with Nb and Hf are sensitive to oxygen and carbon content. The carbon content measured by Atom Probe Tomography (APT) is about 150ppm (precision ± 0.001ppm). No tungsten (from the W electrode) was detected with APT. The oxygen content was not measured. No Mo was detected with APT. Zr and Ta should have appeared in the same mass position than Hf but cannot be seen as the Hf APT pic is stronger.

No specific heat treatment was performed after arc-melting. This alloy was conceived as a demonstrator for the achievement of crystalline coherency and for a validation of the reasoning approach for that purpose. The goal of this work was first to observe the structure in that state even it may be far from any stabilized structure and then, to define, in a second step, heat-treatment to promote precipitation growth or stabilization. If crystalline coherency is observed with small precipitates, it can be lost during their growth. The information about the level of crystal matching is a starting point for further studies.”

Possibly additional consideration about the limit of APT on resolution should be mentioned when considering cluster sizes.

A discussion on these aspects is presented in part ‘5’ (Discussion): I report here this portion of the text.

“Alongside with precipitates, a large number of solute rich clusters, less than 2nm in diameter, are identified. In the present work, cluster sizes are accounted for based on the amount of atoms associated to each cluster using the cluster search algorithm. Considering the amount of atoms only does not allow neither a direct evaluation of metric size nor of the morphology of each cluster. This information could be easily derived using some other algorithms. Nevertheless, insofar as cluster size is in the range of 1 nm, such information is actually deteriorated by the anisotropic spatial resolution of APT and may not reflect the actual size of clusters present in the material. It should also be mentioned that the detector efficiency of the APT used in this study is about 37%. This means that, in average, 37 % of atoms are detected whatever species are considered. Consequently, the amount of atoms associated to each cluster is way underestimated if one wants to get a direct comparison to the amount of atoms per cluster in the material.”

A sentence was modified in the text about APT (part: Atom Probe characterization) in order to avoid some repetitive consideration on this topic.

Discussion is extremely short compared to introduction. The introduction itself contain a lot of information about possible phases, list of binary phases etc, that are not later considered in the discussion or result presentation. The content of the paper is good, but I suggest a re-writing in order to better relate the content of the introduction to the experimental part of the work.

The introduction was reduced to the question/advantage of crystalline coherency and report the design of the alloy in a following part introduced by a global description of the reasoning in order to help the reader to understand the approach. The approach followed was needed for targeting the composition that could allow crystalline coherency.

The text at the end of the introduction focused on coherency is now:

“From the preceding points, it appears that crystalline coherency between the precipitates and the encasing matrix is an essential structural aspect for reaching high temperature mechanical resistance for Nb base superalloys.

The first part of the paper is dedicated to the presentation of the reasoning based on various data in order to reach the chemical composition for the design of an alloy with crystalline coherency between the precipitates and the matrix. In a second part, a Nb-Hf-Ni ternary alloy prepared by arc-melting was characterized with Transmission Electron Microscopy and Atom Probe Tomography, showing that crystalline coherency can be achieved even the initial Nb-Ni system does not seem to support this possibility.”

The part about the design of the alloy is in a part ‘Method: alloy design towards the achievement of coherency’ and this part starts with a general description of the approach in order to help the reader to understand the reasoning. This text is now:

“Method: alloy design towards the achievement of coherency

This part is dedicated to the design description of an alloy showing crystalline coherency. The reasoning for reducing the trial/error experimental work starts from the description of the chemicals necessary for the achievement of a precipitate-matrix structure referring to a work where the question of crystalline coherency was not addressed. As Ni was proposed for the induction of precipitation, the Nb-Ni binary phase diagram is then described leading to the point that this system does not present crystalline matching between Nb and any referenced Nb-Ni phases. Several aspects are thus presented that can be used for targeting crystalline coherency: one of the solution proposed is to add a third element to Nb and Ni, which enlarges the affinity with Ni: Hf offers a larger potential than Zr. The Nb-Hf-Ni phases are described leading to the idea that there are many phases with various crystallographic symmetry and different number of atoms per cell. Based on these aspects, it is concluded that the Nb-Hf-Ni system presents the potential to create crystalline coherency as there is a large ‘crystalline versatility’ among the different Hf-Ni phases (no Nb-Hf-Ni ternary phases were described in the literature). The last step of the reasoning is then focused on the choice of the composition of the Nb-Hf-Ni ternary alloy that was cast and structurally characterized.”

The rest of the text (of this part) was improved. I kept most of it because this constitutes a large part of the work that was focused on the reduction of the trial/error experimental work in order to converge to an alloy composition with crystalline coherency.

We have enriched the comments and discussions. The legend of the figures are already much commented.

Please pay attention to English language, and editing.

Examples:

- Page 1-Line 25 capital letters for Transmission Electron Microscopy

- P1 L31: …in a Ni based solid-solution matrix, named G.

- P2 L58: remove ‘,’.

- P2 L89; replace ‘synthesis’ with ‘review’

- P4 L161 hR13

- P12 L352, two specimen of the alloys were analysed, with different size.

- P12 L358: no specific heat treatment was performed, so avoid reference to an applied heat treatment’.

- P12 L354: A sentence is present twice. (‘Meanwhile…’).

- P17 L540 Crystalline…

These aspects were corrected directly in the text, and a careful reading was achieved regarding possible similarities (an English colleague read the text for removing them).

It must be added that the text was improved ‘here and there’ and these modifications are in the second version of the draft that is already available.

I am open to any further comments that can improve the text: many thanks for your reading and stimulating comments.

Reviewer 2 Report

The submission is concerned with the development of Niobium based supperalloys. Overall the subject of the paper is worth of investigation. The results are ample, various analytical techniques have been used, the discussion is apt and thus overall the manuscript  presents itself well and I would recommend it for publication, provided some major criticism is addressed beforehand. Firstly the paper is ill structured and organised. There is a stark imbalance between the introductory part and the actual contributory part of the submission. The equilibrium has to be restored before the paper is given any further more detailed attention:

1.      There is no need to provide literature reference related to any previous work in the abstract. Briefly the main motivation, work carried out and main results should be highlighted.

2.      The introduction part is way, way (!!!) to lengthy. It reads almost like a book chapter or a review article. Too many aspects are brought forth, and what makes it even worse it is done frequently in a confusing and ill ordered manner. It is very difficult to follow. The authors jump from cross slip mechanisms, to diffusion and then to Hume Rothery. Many of this is text book knowledge.  Please rewrite the introduction by stating the most pertinent state of the art and then motivation for work undertaken and presented. Otherwise you will scare potential readers.

3.      The Materials and Methods contains too any irrelevant details e.g. on sample mounting to a micromanipulator needle or sentences like “the thinning was performed until a small hole was obtained in the 287 middle of the 3mm diameter disk…”. Most potential readers will likely have at least a faint idea of sample preparation for TEM if not they can always look up more specific literature or internet. In this context it just takes up space and makes the paper unnecessarily too long and heavy. It took the authors 7 pages! before they even got to the results section. To be frank personally I already have been discouraged from further reading. Think of the busy schedules and very often limited time people have at their disposal these days!

4.      In the results and discussion part please try harder to describe what you see and what is relevant and discuss rather than merely quote the reference.

5.      Put the figures in a more compact, concise and logical manner e.g. put together any relevant or corresponding BF and SADP etc. The pictures and randomly isolated what takes up space and consumes too many pages. It is difficult to browse through.

Please address the following first before any more detailed remarks can be provided.

Author Response

The submission is concerned with the development of Niobium based superalloys. Overall the subject of the paper is worth of investigation. The results are ample, various analytical techniques have been used, the discussion is apt and thus overall the manuscript presents itself well and I would recommend it for publication, provided some major criticism is addressed beforehand. Firstly the paper is ill structured and organised. There is a stark imbalance between the introductory part and the actual contributory part of the submission. The equilibrium has to be restored before the paper is given any further more detailed attention

We have worked out these aspects and reorganised the cutting out of the text with improvement of the text, synthesizing the approach at the beginning of the new part “Method: Alloy design towards the achievement of coherency”.

The introduction was reduced to the question/advantage of crystalline coherency and report the design of the alloy in a following part introduced by a global description of the reasoning in order to help the reader to understand the approach. The approach followed was needed for targeting the composition that could allow crystalline coherency.

The text at the end of the introduction focused on coherency is now:

“From the preceding points, it appears that crystalline coherency between the precipitates and the encasing matrix is an essential structural aspect for reaching high temperature mechanical resistance for Nb base superalloys.

The first part of the paper is dedicated to the presentation of the reasoning based on various data in order to reach the chemical composition for the design of an alloy with crystalline coherency between the precipitates and the matrix. In a second part, a Nb-Hf-Ni ternary alloy prepared by arc-melting was characterized with Transmission Electron Microscopy and Atom Probe Tomography, showing that crystalline coherency can be achieved even the initial Nb-Ni system does not seem to support this possibility.”

The part about the design of the alloy is in a part ‘Method: alloy design towards the achievement of coherency’ and this part starts with a general description of the approach in order to help the reader to understand the reasoning. This text is now:

“Method: alloy design towards the achievement of coherency

This part is dedicated to the design description of an alloy showing crystalline coherency. The reasoning for reducing the trial/error experimental work starts from the description of the chemicals necessary for the achievement of a precipitate-matrix structure referring to a work where the question of crystalline coherency was not addressed. As Ni was proposed for the induction of precipitation, the Nb-Ni binary phase diagram is then described leading to the point that this system does not present crystalline matching between Nb and any referenced Nb-Ni phases. Several aspects are thus presented that can be used for targeting crystalline coherency: one of the solution proposed is to add a third element to Nb and Ni, which enlarges the affinity with Ni: Hf offers a larger potential than Zr. The Nb-Hf-Ni phases are described leading to the idea that there are many phases with various crystallographic symmetry and different number of atoms per cell. Based on these aspects, it is concluded that the Nb-Hf-Ni system presents the potential to create crystalline coherency as there is a large ‘crystalline versatility’ among the different Hf-Ni phases (no Nb-Hf-Ni ternary phases were described in the literature). The last step of the reasoning is then focused on the choice of the composition of the Nb-Hf-Ni ternary alloy that was cast and structurally characterized.”

The rest of the text (of this part) was improved. I kept most of it because this constitutes a large part of the work that was focused on the reduction of the trial/error experimental work in order to converge to an alloy composition with crystalline coherency.

There is no need to provide literature reference related to any previous work in the abstract. Briefly the main motivation, work carried out and main results should be highlighted.

I have corrected this point and rewritten the abstract partly with some additional points to make the approach clearer.

The abstract is now:

“In a previous work, the chemical elements necessary for the achievement of Niobium base superalloys were defined in order to get a structure equivalent to that of Nickel base superalloys, which contain ordered precipitates within a disordered solid-solution. It was especially emphasized that precipitation hardening in the Niobium matrix would be possible with the addition of Ni. The remaining question about the design of such Niobium superalloys concerned the achievement of ordered precipitates in crystalline coherence with the Nb matrix i.e. with a crystalline structure equivalent to the Nb crystal prototype and with a lattice parameter in coherency with that of the Nb matrix. In order to reduce the trial/error experimental work, a reasoning based on various data for the achievement of coherency is presented. Then, starting from the Nb-Hf-Ni ternary alloy thus defined, this paper demonstrates that the precipitation of an ordered Nb phase within a disordered Nb matrix can be achieved with lattice parameter coherency between the ordered precipitates and the disordered matrix. The chemistry and the crystallographic structure of the precipitates were characterized using Transmission Electron Microscopy and Atom Probe Tomography. These results can help to conceive a new family of Nb base superalloys.”

2. The introduction part is way, way (!!!) to lengthy. It reads almost like a book chapter or a review article. Too many aspects are brought forth, and what makes it even worse it is done frequently in a confusing and ill ordered manner. It is very difficult to follow. The authors jump from cross slip mechanisms, to diffusion and then to Hume-Rothery. Many of this is textbook knowledge. Please rewrite the introduction by stating the most pertinent state of the art and then motivation for work undertaken and presented. Otherwise you will scare potential readers.

I have reduced the introduction to the question/advantage of crystalline coherency and report the design of the alloy in a following part introduced by a global description of the reasoning in order to help the reader to understand the approach. The approach followed was needed for targeting the composition that could allow crystalline coherency. I thought that recalling some basic knowledge could reduce the time of the reader, avoiding him the necessity to jump from a reference (to be found …) to another if he needed to have a good understanding of the reasoning or, if that knowledge had partly evaporated from his memory.

3. The Materials and Methods contains too any irrelevant details e.g. on sample mounting to a micromanipulator needle or sentences like “the thinning was performed until a small hole was obtained in the (p287) middle of the 3mm diameter disk…”. Most potential readers will likely have at least a faint idea of sample preparation for TEM if not they can always look up more specific literature or internet

The text of this part is now: Materials and experimental methods

About 300g of the Nb-Hf-Ni ternary alloy was arc-melted with a tungsten electrode under an argon atmosphere. The purity was better than 99at% for Nb (the main impurity is Ta), better that 97at% for Hf (the main impurity is Zr, typically about 2.5at%), and better than 99.5at% for Ni. The secondary impurities in Nb and Hf are oxygen, carbon and refractory elements such as Mo, W, Ti (typically below few hundreds of ppm) and Ta also in Hf. It should be pointed out that regarding the electronegativity difference between on the one hand, Nb, Hf and their impurities (Ta, Zr, W, Mo, Ti) and on the other hand, Hf has the largest electronegativity difference (or affinity) with Ni. Thus, even the refractory impurities can represent a few percent, Ni should form mostly precipitates with Hf. Moreover, at these concentration levels, these refractory elements remain in solid-solution: they can also participate to the crystalline matching between the precipitates and the matrix. Even the purity of Hf is the lowest compared to Nb and Ni, considering a Zr amount of 2.8at% in Hf, the Zr content introduced in the alloy with 5.8at%Hf is about 0.16at%. The quantity of the other impurities brought into the alloy with 5.8at% Hf is rather very small. Tantalum which is the major impurity in Nb, has chemical and structural properties very close to Nb and should not have a strong impact on the precipitate structure (Ta is in solid-solution).

After arc-melting, the cooling step of the alloy was not registered: but a gentle cooling was performed allowing the precipitation. The composition of the alloy is: 90.3 at% Nb, 5.8 at% Hf, 3.9 at% Ni (or 86.9 wt% Nb, 10.7 wt% Hf, 2.4 wt% Ni). The properties and phase structures with Nb and Hf are sensitive to oxygen and carbon content. The carbon content measured by Atom Probe Tomography (APT) is about 150ppm (precision ± 0.001ppm). No tungsten (from the W electrode) was detected with APT. The oxygen content was not measured. No Mo was detected with APT. Zr and Ta should have appeared in the same mass position than Hf but cannot be seen as the Hf APT pic is stronger.

No specific heat treatment was performed after arc-melting. This alloy was conceived as a demonstrator for the achievement of crystalline coherency and for a validation of the reasoning approach for that purpose. The goal of this work was first to observe the structure in that state even it may be far from any stabilized structure and then, to define, in a second step, heat-treatment to promote precipitation growth or stabilization. If crystalline coherency is observed with small precipitates, it can be lost during their growth. The information about the level of crystal matching is a starting point for further studies.

3.1. Specimen preparation by Focussed Ion Beam (FIB) milling

The specimens were prepared using the standard lift-out and mounting method [58] using a Plasma Focused Ion Beam “Helios G4 PFIB CXe” (Thermo Fisher Scientific). A sample was first extracted from the Nb-Hf-Ni substrate by ion milling, using 30 kV Xe+ ions. It was welded to a micromanipulator using ion beam induced deposition of platinum, and sections of the sample were welded in the same way on tungsten needles. These specimens were subsequently sharpened with the Xe+ ion beam, using the annular milling method [58]. In order to minimize the thickness of the final damaged surface layer, the needles were then milled over ~100 nm using a 12 kV Xe+ ion beam. Using this method, Nb-Hf-Ni needles with a radius of curvature below 50 nm could be obtained (Figure 1).

Figure 1. Scanning Electron Microscopy image of a Nb-Hf-Ni specimen after annular milling by FIB.

3.2. Atom Probe Tomography

Samples were analysed by Atom Probe Tomography (APT). This technique was selected because of its ability to analyse the chemistry and the morphology of precipitates at the atomic level. The experiments were carried out on a LEAP 4000 HR device from CAMECA. The experiments were performed at 60K with a flux of 0.25 ions/S/nm². The reconstruction procedure and analysis were conducted with the software package IVAS®. The cluster search algorithm, which is the maximum separation distance algorithm, was applied based on the distribution of 8th nearest neighbours distances. A maximum separation distance of 0.8 nm was identified as relevant according to the comparison of experimental data to randomized data. A minimum amount of 35 solute atoms in clusters was set to remove all artificial clusters eventually detected in the solid solution. The reader can find more details about the methodology in reference [58].

3.3. Transmission Electron Microscopy

High Resolution Transmission Electron Microscopy (HR-TEM), Scanning Transmission Electron Microscopy (STEM) and Energy Dispersive X-Rays Spectrum (EDX) for elemental mapping, were carried out using a TECNAI OSIRIS (Thermo Fisher Scientific) transmission electron microscope operated at 200kV, equipped with a 4K GATAN camera.

The thin foil for TEM observation was prepared using classical mechanical polishing and ionic thinning machine. The thickness of the observed region is typically of several tenths of nm. Several thin foils were extracted from different parts of the ingot and showed similar TEM/STEM-EDX results about the structure and the chemistry of the precipitates. No other means were used to check homogeneity.

In this context it just takes up space and makes the paper unnecessarily too long and heavy. It took the authors 7 pages! before they even got to the results section. To be frank personally I already have been discouraged from further reading. Think of the busy schedules and very often limited time people have at their disposal these days !

See the answer above. With the structure proposed in the second draft, the reader can jump directly from the introduction to the experimental results if he wishes to avoid the ‘Alloy design part’.

4. In the results and discussion part please try harder to describe what you see and what is relevant and discuss rather than merely quote the reference.

We have enriched the comments and discussions. The legend of the figures are already much commented.

5. Put the figures in a more compact, concise and logical manner e.g. put together any relevant or corresponding BF and SADP etc. The pictures and randomly isolated what takes up space and consumes too many pages. It is difficult to browse through.

I have reorganized the figures and reduced them. I kept figure 2 (with dislocations) with a rather large size because this an old picture (scanned) and dislocation pinning might be difficult to see with a smaller size.

Please address the following first before any more detailed remarks can be provided.

It must be added that the text was improved ‘here and there’ and these modifications are in the second version of the draft that is already available.

I am open to any further comments that can improve the text: many thanks for your reading and stimulating comments.

Round 2

Reviewer 2 Report

The manuscript appears to have been improved. The authors have provided detailed answers to the criticism raised. Although I still think the manuscript is unnecessarily too long it can be published.